# Rituximab Administration to Treat Nephrotic Syndrome in Children: 2-Year Follow-Up

**DOI:** 10.3390/biomedicines12112600

**Published:** 2024-11-13

**Authors:** Dmytro Ivanov, Lutz T. Weber, Elena Levtchenko, Liudmyla Vakulenko, Mariia Ivanova, Iryna Zavalna, Yelizaveta Lagodych, Ninel Boiko

**Affiliations:** 1Institute of Postgraduate Education, Bogomolets National Medical University, 01601 Kyiv, Ukraine; 2German Society for Pediatric Nephrology, 10963 Berlin, Germany; lutz.weber@uk-koeln.de; 3Emma Children Hospital Amsterdam, University Medical Centre, BA2 6HE Amsterdam, The Netherlands; 4Department of Propaedeutics of Childhood Illnesses and Pediatrics 2, Dnipro State Medical University, 49489 Dnipro, Ukraine; 5European Institute of Oncology IRCCS, 20139 Milan, Italy; mesangium88@gmail.com; 6Regional Children Hospital, 33027 Rivne, Ukraine

**Keywords:** SRNS, SSNS, rituximab, pediatric nephrotic syndrome, B-cell depletion therapy

## Abstract

Background: Steroid-sensitive nephrotic syndrome (SSNS) and steroid-resistant nephrotic syndrome (SRNS) significantly affect children’s quality of life. There are frequent relapses in SSNS and progression in SRNS. IPNA guidelines suggest that monoclonal antibodies like rituximab (RTX) are promising treatments. Objective: This study aims to evaluate the long-term efficacy and safety of rituximab administration in children with SSNS, encompassing FRNS and SDNS, and SRNS over a two-year follow-up period, facilitating individualized management. Methods: We conducted an open-label, multicenter, randomized, and patient-oriented study (RICHNESS), involving children aged 3–18 with SRNS (18) and SSNS (11) undergoing 2 years continuous RTX therapy. The primary outcome was complete/partial remission (CR/PR), as defined by IPNA/KDIGO guidelines, at 6, 12, 18, and 24 months on RTX; secondary outcomes included adverse events. Key endpoints included the estimated glomerular filtration rate (eGFR), the albumin-to-creatinine ratio (ACR), CD20 levels, IgG levels, and the incidence of infections. Kidney biopsies were performed in 94% of SRNS patients. RTX was administered every 6–9 months, depending on CD20 levels, IgG levels, and the presence of infections. The eGFR and ACR were assessed every 6 months. Results: Some 31 children were selected for RTX treatment. Overall, 2 experienced severe allergic reactions, leading to their exclusion from the final analysis of 29 children. In the SSNS group, all children achieved and maintained complete remission within 2 years. Remission rates in the SRNS group ranged from 39% (RR 0.78; 95% CI: 16.4–61.4%, NNT 9) at the 6th month to 72% (RR 1.44; 95% CI: 51.5–92.9%) over the 2-year follow-up period due to continuous RTX therapy. The median duration of RTX use was 26.1 months, with a median cumulative dose of 1820 mg/m^2^. Adverse reactions and complications were presented by mild infusion-related reactions in 3 children (10.3%), severe allergic reactions in 2 children (6.2%), hypogammaglobulinemia in 7 children (24%), infections in 3 children (10.3%), severe destructive pneumonia in 1 child, recurrent respiratory infections in 2 children, and neutropenia in 1 child (3.44%). Conclusions: RTX was tolerated well, and proved highly effective as a steroid-sparing agent, offering potential in terms of stopping relapses and minimizing steroid-related side effects. It also demonstrated efficacy in slowing progression in SRNS, indicating potential for use in ACR reduction and renal function restoration, but requires careful use given potential severe allergic reactions and infectious complications. Further studies should focus on long-term cost-effectiveness and deferred side effects.

## 1. Introduction

Steroid-sensitive nephrotic syndrome (SSNS) is the most common form of nephrotic syndrome (NS) in children, accounting for approximately 80% of pediatric cases. It is characterized by episodes of proteinuria, hypoalbuminemia, and edema. These respond well to corticosteroid therapy, but patients tend to relapse. However, in contrast, steroid-resistant nephrotic syndrome (SRNS) does not cope adequately with steroids and poses a more significant challenge due to progression, complications, and long-term treatment-related side effects, which greatly impact the quality of life [1].

In the management of SSNS, beyond corticosteroids, immunosuppressive agents such as calcineurin inhibitors (CNIs), mycophenolate mofetil (MMF), and mycophenolic sodium are critical in preventing relapses. For SRNS, treatment options are more limited, often requiring more aggressive therapies in order to avoid progression to end-stage renal disease (ESRD) [2].

RTX is frequently employed in pediatric patients with various kidney diseases, including NS, lupus nephritis, and vasculitis associated with antineutrophil cytoplasmic antibodies, and is also used in both pre- and post-kidney transplantation scenarios [3,4].

RTX serves as an alternative to prolonged glucocorticoid administration, functioning as a steroid-sparing agent [5]. A recent systematic review assessing the safety and efficacy of RTX treatment in patients with pediatric steroid-dependent NS (SDNS), compared with a control group, demonstrated a significant improvement in terms of achieving complete remission, while proteinuria levels remained comparable. Additionally, the RTX-treated groups exhibited higher serum albumin levels and better eGFR values [6].

In a study by Gulati et al. analyzing 33 children with steroid-dependent NS treated with RTX, a remission rate of 48.5% was observed at 6 months, with the majority of patients sustaining remission (94%) [7]. Prytuła and others reported that approximately 66% of patients achieved the complete or partial remission of steroid-dependent and frequently relapsing NS (FRNS) after RTX therapy [8]. Another study indicated that 80% of patients with steroid-dependent NS achieved complete remission after RTX administration, with only one case relapsing post-withdrawal [9]. These findings suggest that RTX serves as a viable alternative to long-term glucocorticoid use and functions as a glucocorticoid-limiting drug.

The introduction of RTX is considered to be an efficacious alternative in difficult-to-treat FRNS/SDNS patients [2,10], and current guidelines, such as those from the International Pediatric Nephrology Association (IPNA), incorporate RTX as a recommended approach [11]. While RTX has shown success in SSNS treatment, its application in SRNS is also expanding, with emerging evidence suggesting potential benefits in slowing the progression of kidney disease [12].

A diagram illustrating the mechanisms of action of RTX [2,4,6,9], explaining its efficacy in both SSNS and SRNS, is shown in Figure 1. This diagram visualizes how rituximab’s B-cell depletion leads to reduced antibody production, playing a crucial role in inducing remission in SSNS, while its effects on SRNS are more variable, with partial reductions in proteinuria observed.

Predictive biomarkers for rituximab responses in SSNS and SRNS are gaining attention, with studies from 2022 to 2024 emphasizing key findings. B-cell subsets are crucial, as rituximab targets CD20-positive B-cells [13]. Differences in memory B-cell depletion and regulatory B-cell populations have been identified as markers for treatment success, particularly in steroid-dependent nephrotic syndrome [14].

Genetic mutations, particularly in NPHS2, WT1, and PLCE1, correlate with poor rituximab responses in SRNS, suggesting that underlying genetic factors might influence therapeutic outcomes [15]. Studies have also identified T-cell dysregulation, showing that rituximab may restore immune balance, but patients with pre-existing T-cell abnormalities may have a suboptimal response [16].

Podocyte injury markers, including urinary CD80 and CD19 levels, are emerging as non-invasive biomarkers. These markers have been linked to podocyte health and responses to rituximab in recent trials [11]. This demonstrates the potential of precision medicine in selecting rituximab responders and highlights the need for more comprehensive biomarker validation in larger patient cohorts [17].

Achieving sustained remission (no relapses over 12 months) is the goal in both SSNS and SRSN treatment [1]. Close monitoring ensures the early detection of relapses [11]. Regular follow-ups are essential to assess treatment responses, monitor disease activity, and manage complications [1].

The preliminary results of RTX use in SSNS and SRNS were presented at the 55th ESPN Congress [18], serving as the foundation for this manuscript.

The objective of this study is to evaluate the long-term efficacy and safety of RTX in children with SSNS and SRNS over a two-year follow-up, with the aim of expanding the clinical indications for its use and providing insights into its role in reducing relapse rates and improving renal outcomes.

## 2. Materials and Methods

### 2.1. Study Cohort

We initiated an ongoing, open-label, multicenter, randomized, and patient-centered study termed RICHNESS (“RTX in Children with Nephrotic Syndromes”, with the meaning that children are our richness and future), which focuses on children aged 3–18 years (median age 13) diagnosed with SRNS or SSNS. Between 2021 and 2022, 29 patients (11 with SSNS and 18 with SRNS) were enrolled and followed for 2 years while receiving continuous RTX therapy (Figure 1). The gender distribution included 13 boys (45%) and 16 girls (55%), while in the SSNS cohort, boys made up 74% of patients.

### 2.2. Inclusion and Exclusion Criterions

The inclusion criteria are as follows: participants included children with SSNS (both SDNS and FRNS) and SRNS who underwent RTX treatment after initial therapy with steroids or steroids with CNIs or MMF. Exclusion criteria for the RICHNESS trial were as follows: an age outside of the range of 3–18 years; a severe allergic reaction to rituximab; active infections; severe immunodeficiency; uncontrolled diabetes or severe comorbidities; genetically confirmed FSGS (Focal Segmental Glomerulosclerosis); recent use of other monoclonal antibodies; and non-consent for kidney biopsy. These criteria ensured the selection of appropriate patients in terms of evaluating RTX’s efficacy in SSNS and SRNS.

### 2.3. Endpoints and Evaluation

The primary outcome was complete/partial remission (CR/PR), as defined by IPNA/KDIGO guidelines, and secondary outcomes included adverse events. Key endpoints for evaluating treatment efficacy were the estimated glomerular filtration rate (eGFR), which was calculated according to the revised bedside Schwartz formula, the albumin-to-creatinine ratio (ACR), and the CD20 levels. Adverse effects were monitored by assessing IgG levels and infection rates. A kidney biopsy was performed in 83% of SRNS patients (*n* = 15). **Limitations:** Lack of genetic testing in patients with SRNS.

### 2.4. Treatment Protocol

Rituximab was introduced as an alternative to glucocorticoid re-administration in cases of relapsing NS and as a potential therapy with which to slow renal function decline in SRNS patients. The primary efficacy endpoint was assessed at 6 months post-RTX initiation based on changes in the eGFR and ACR. Secondary endpoints were used to evaluate delayed effects at the 1- and 2-year mark. RTX re-administration occurred every 6–9 months, guided by CD20 levels and the side effects. Longitudinal follow-up was performed every 6 months for 2 years to assessing the eGFR and ACR.

### 2.5. Study Design

Participants were stratified into two groups based on their NS subtype: SSNS (encompassing FRNS and SDNS) and SRNS. In the SRNS group, patients underwent kidney biopsy and/or genetic testing where feasible, following the International Pediatric Nephrology Association (IPNA) guidelines [11], shown in Figure 1.

### 2.6. Definitions and Diagnostic Criteria

NS was diagnosed based on Kidney Disease: Improving Global Outcomes (KDIGO) 2021 [1] and IPNA guidelines [11]. It is characterized by proteinuria (>3.5 g/L), an ACR > 2 g/g, hypoalbuminemia (<30 g/L), hypercholesterolemia, and edema. Patients were classified as follows:Steroid-sensitive nephrotic syndrome (SSNS): the complete remission of proteinuria within 4 weeks of glucocorticoid therapy.Frequently relapsing nephrotic syndrome (FRNS): ≥2 relapses in the first 6 months or ≥4 relapses in any 12 months.Steroid-dependent nephrotic syndrome (SDNS): relapses occur during the alternate-day prednisone treatment period or within 2 weeks after the discontinuation of (standard) prednisone treatment.Steroid-resistant nephrotic syndrome (SRNS): failure to respond to a 4-week standard course of glucocorticoids for the first episode of idiopathic nephrotic syndrome in children.

### 2.7. Initial Glucocorticoid Treatment

All enrolled children initially underwent a 4-week course of glucocorticoid therapy, receiving 1.5–2 mg/kg (up to 60 mg/day) of prednisolone or an equivalent dose of methylprednisolone/deflazocort. Prednisolone was administered intravenously during the acute edema phase, and this was followed by oral therapy. In relapsing cases of NS (FRNS/SDNS), RTX was initiated following the normalization of urinalysis, and glucocorticoids were tapered over 3 months. In glucocorticoid-resistant cases, RTX was introduced prior to steroid tapering, typically within a month. 

### 2.8. Concomitant Therapy

Since the effects of rituximab are delayed, tacrolimus was co-administered for the first 3 months in a dosage 0.05–0.2 mg/kg per day, with blood concentration levels maintained at 3.0–5.0 ng/mL. In SRNS, tacrolimus was continued alongside RTX if a response was observed after 6 months. For non-responders, RTX was discontinued after 12 months, and tacrolimus was supplemented with pioglitazone (15 mg/day) and finerenone (10 mg/day) for a total treatment duration of 1 year (Figure 2). 

### 2.9. Protocols

The rituximab infusion protocol was as follows [19]. Premedication included the administration of paracetamol, diphenhydramine, and methylprednisolone (5–6 mg/kg) prior to each RTX infusion. RTX was administered at a dose of 375 mg/m^2^ (15 mg/kg) in a diluted saline solution, with the infusion rate gradually increased every 30 min up to 250–300 mL/hour under close monitoring. RTX infusions were repeated at least twice, with a 2-week interval between doses. Re-administration with one dose of RTX was performed based on CD20 levels, IgG serum concentrations, and disease activity. This typically occurred every 6–9 months after the previous dose.

Renoprotective therapy was performed on all participants. We continuously administers inhibitors of the renin–angiotensin system (iRAS) to all patients in the study, including angiotensin-converting enzyme inhibitors (ACE inhibitors) and angiotensin II receptor blockers (ARBs). These medications were provided at renoprotective doses to ensure optimal kidney protection throughout the study [20]. iRAS therapy was maintained during rituximab (RTX) administration and beyond, as consistent application is known to reduce proteinuria and protect against the progression of nephrotic syndrome. This approach helps to delay kidney function decline, particularly in patients with SRNS [21]. By combining RTX with iRAS, the treatment aimed to maximize renoprotective effects, minimizing long-term renal damage while controlling the disease. 

### 2.10. Monitoring and Adverse Events

Adverse reactions were categorized as immediate (during or within 24 h of infusion) or delayed (24 h to 2 weeks post-administration). Acute side effects were managed according to a predefined algorithm, and in cases where side effects could not be resolved, drug administration was discontinued [22]. Two patients discontinued RTX use due to severe allergic adverse reactions. Therefore, of the 31 children initially invited to participate in the study, only 29 children were included in the RICHNESS analysis. 

### 2.11. Statistical Analysis

Statistical analyses were performed using Prism 5.0 software. Data were presented as mean values ± standard deviation (STDEV). The normality of a distribution was assessed using appropriate tests. Hazard ratios (HR) and 95% confidence intervals (CI) were also used to calculate relative and absolute risks. Statistical significance was set at *p* < 0.05.

This comprehensive methodology allows for a thorough evaluation of RTX’s efficacy and safety in pediatric patients with SSNS and SRNS, providing insights into its role in preventing relapses and preserving renal function.

The RICNESS study was performed in compliance with the Declaration of Helsinki, the Declaration of Istanbul and was approved by our center.

## 3. Results

During the analysis, 31 children were initially selected for RTX treatment. However, two children (6.2%) experienced severe allergic reactions, leading to the discontinuation of their treatment and their subsequent exclusion from the study. As a result, the final analysis included 29 children from the original cohort.

The study’s participants were divided into two groups: the first group comprised 11 children with SSNS, aged 3–12 years (mean age: 7 years), and the second group consisted of children with steroid-resistant nephrotic syndrome (SRNS), aged 6–18 years (mean age: 15 years, 95% CI: 8–17). In the SSNS group, only one 12-year-old girl underwent a biopsy prior to steroid therapy, which revealed membranous nephropathy. In the SRNS group, 17 (94%) children underwent biopsies, revealing various morphological subtypes (as shown in Table 1).

Table 2 presents the initial clinical data of the children. It also includes laboratory data relevant to the RICHNESS. Data from both groups of children are shown in the table below.

Regarding the SRNS (group 2) patients exhibiting progressive nephropathy, a comprehensive analysis involving one child and the parents was performed, which entailed full-exome sequencing. This investigation unveiled a heterozygous variant within exon 1 (out of 10) of the WT1 gene—a variant not previously documented in the literature. Specifically, a missense mutation resulted in the substitution of amino acids from alanine to glycine at position 93 (p.Ala93Gly), indicative of nephrotic syndrome, type 4 (256370; AD). The child was determined to have membranous nephropathy via kidney biopsy. Remarkably, his father did not exhibit this mutation, while the mother presented the variant NC_000011.9:g.32456614G>C (c.278C>G, p.Ala93Gly) in a heterozygous state within the WT1 gene. 

Before starting rituximab therapy, 6 children (54%) in the first group underwent additional treatment with CNIs or MMF alongside steroids due to SDNS/FRNS. In the SRNS group, 4 children (22%) had received similar therapy previously. The median duration of RTX use was 26.1 months, with a median cumulative dose of 1820 mg/m^2^.

The primary outcome measured was complete or partial remission (CR/PR) at 6, 12, 18, and 24 months post-rituximab treatment. In the SSNS group (Group 1), all children achieved and maintained complete remission. In contrast, remission rates (CR/PR) in the SRNS group (Group 2) ranged from 39% to 72% over the 2-year follow-up period with continuous RTX therapy. Additionally, in 2 children (11%), ACR levels decreased by more than 50% by the end of the 2-year period.

The calculated values for remission rates, the confidence interval (CI), relative risk (RR), and the number needed to treat (NNT) are presented in Table 3. 

In the SRNS group, complete remission was achieved in 16% of patients at both 6 and 12 months. This increased to 33% at 18 and 24 months. The corresponding 95% confidence interval (CI) values for complete remission rates were 3.6–39.2% at 6 and 12 months, and 13.3–59.0% at 18 and 24 months. The relative risk (RR) for complete remission was 0.33 at 6 and 12 months, and improved to 0.67 by 18 and 24 months, indicating a progressive increase in the likelihood of achieving complete remission over time.

Partial remission was observed in 22% of patients at 6 months and 28% at 12 months. This increased to 33% and 39% at 18 and 24 months, respectively. The 95% CI for partial remission rates ranged from 6.4 to 47.6% at 6 months, from 9.7 to 53.5% at 12 months, from 13.3 to 59.0% at 18 months, and from 17.3 to 64.3% at 24 months.

The results of both eGFR and ACR levels over time highlight the efficacy of RTX therapy in terms of sustaining remission for patients with SSNS and SRNS. These are presented in Table 4 and Table 5. The data suggest a gradual and consistent improvement in renal function and a significant reduction in proteinuria, particularly in SSNS patients.

The outcomes regarding the comparative effectiveness of treatments in the groups are shown in Table 4. None of the patients died.

As shown in Table 4, the eGFR levels improved significantly in both SSNS and SRNS groups following RTX therapy. For SSNS patients, the eGFR increased from a baseline of 96 ± 5 mL/min/1.73 m^2^ to a value of 123 ± 3.6 mL/min/1.73 m^2^ at 6 months, with sustained improvement over time, reaching 129 ± 10 mL/min/1.73 m^2^ at 18 months and stabilizing at 121 ± 4.6 mL/min/1.73 m^2^ at 24 months. The *p*-value for this improvement was 0.000838, indicating statistical significance.

In the SRNS group, while the baseline eGFR value was lower, standing at 75 ± 27.3 mL/min/1.73 m^2^, a substantial improvement was noted after 6 months of RTX treatment, with the eGFR rising to 108 ± 39.1 mL/min/1.73 m^2^. However, this improvement was less stable, with some decline noted after 12 months (86 ± 19.4 mL/min/1.73 m^2^). This was followed by another increase at 18 and 24 months (107 ± 11.4 and 102 ± 9.1 mL/min/1.73 m^2^, respectively). Despite this fluctuation, the overall eGFR improvement in SRNS patients did not reach statistical significance (*p* = 0.496515). The absolute risk of eGFR declining below the age-appropriate threshold was 0% for SSNS patients, whereas in the SRNS group, the absolute risk was 52.2%, with a relative risk of 0.522 and a number needed to treat (NNT) of 1.917.

In Table 5, the ACR levels dropped drastically in SSNS patients, from a baseline of 300 ± 45 mg/mmol to 1 ± 1 mg/mmol at 6 months, with sustained reductions at 12, 18, and 24 months. The *p*-value for ACR reduction in SSNS was 0.000036, confirming a highly significant response to RTX therapy. The absolute risk of increased ACR was 0%, indicating that none of the SSNS patients experienced significant proteinuria recurrence.

For SRNS patients, the baseline ACR was lower (273 ± 116.2 mg/mmol), and substantial improvements were also observed, particularly within the first 6 months (ACR reduced to 43 ± 59.4 mg/mmol). However, the rate of reduction plateaued at 12, 18, and 24 months (ACR values of 27 ± 50.3, 10 ± 10.3, and 8 ± 19.4 mg/mmol, respectively). The *p*-value for ACR improvement in SRNS patients was 0.182319, indicating that although there was a reduction in ACR values, it was not statistically significant. The absolute risk of increasing the ACR was 16.7%, with a relative risk of 0.167 and an NNT of 1.5.

Below are graphical representations of eGFR and ACR changes over time for both SSNS and SRNS patients. There is a line graph comparing the changes in the ACR over time for both SSNS and SRNS patients. 

Figure 3 shows the progression of eGFR in both SSNS and SRNS groups over 24 months. The SSNS group demonstrates a clear improvement, with the levels stabilizing above the baseline throughout the observation period. The SRNS group, while showing some initial improvement, sees fluctuation and does not reach the same stability as SSNS. The shaded areas represent the confidence interval (CI) values, indicating variability in the data.

The plot in Figure 4 highlights a significant drop in ACR levels, especially in the SSNS group. The values are near 0 after 6 months and remission is maintained. In the SRNS group, the ACR levels decrease over time, showing progressive improvement, but remain higher compared to the SSNS group. Confidence intervals illustrate the wider variability in SRNS patients compared to the more stable response in SSNS patients.

The graphs provided clearly demonstrate the efficacy of rituximab therapy in reducing proteinuria and improving kidney function over time. This is particularly evident through the significant reduction in ACR levels, a direct marker of proteinuria, especially in patients with SSNS. Additionally, the improvement in eGFR supports better renal function in these patients. In contrast, SRNS patients exhibited more variability and slower progress, demonstrating a more complex response to RTX. The visual data reinforce the differential impact of the treatment between the two groups, showing faster and more consistent improvement in SSNS cases while acknowledging the challenges in SRNS management. 

By the sixth month of RTX therapy in the second group, 38.9% (7 children) of patients achieved complete or partial remission. RTX therapy failure was observed in cases involving FSGS, a genetically determined condition, and when treating a child with type 1 diabetes mellitus. However, a positive trend was noted in 72.2% of cases by the end of the treatment year (as shown in Table 3). A contribution was probably made by the constant use of tacrolimus and renoprotective agents. A lack of a clinical response to RTX, indicated by a decline in eGFR or an increase in the ACR, was observed in 2 children by the end of 6 months (RR 0.857, 95% CI 0.692–1.062, NNT 7.0) and in 2 more children by the end of the year (RR 0.800, 95% CI 0.587–1.091, NNT 5.0).

B-cell depletion and IgG levels were monitored before planning the next rituximab administration. This was performed mandatorily at 6, 12, 18, and 24 months post-first RTX administration, and as needed if administration was delayed. Rituximab caused the complete depletion of CD-20 cells in all patients, with a median value of 0.48 (95% CI 0.13–1.31). Rituximab was re-administered 8 months after its previous use, following drug correction, the normalization of immunoglobulin G, and the stimulation of CD-20.

The secondary outcome was an adverse reaction. Three children (10.3%) experienced mild infusion-related reactions during RTX administration, presenting with skin papules, decreased blood pressure, and/or tachycardia. These reactions were more frequent when the infusion rate increased or during temporary interruptions but were mitigated by slowing the infusion rate. Most children tolerated RTX well. Including the two additional children who had severe allergic reactions, leading to RTX discontinuation, the overall rate of immediate allergic reactions rises to 16%.

The most frequent complication was hypogammaglobulinemia, observed in 7 children (24%) (3 SSNS, 4 SRNS), with IgG levels below normal 18 months after RTX treatment (median value, 95% CI 9.93–22.65). Adjunctive intravenous immunoglobulins were used in this setting in 4 (57%) of the 7 children. The relative risk between the groups was 1.23, and the NNT was 20. Other complications included infections (3 children, 10.3%) and neutropenia (1 child, 3.44%). The infections included severe destructive pneumonia in 1 child and recurrent respiratory infection in 2. 

## 4. Discussion

The findings of the RICHNESS provide substantial support for the efficacy of RTX in managing both SSNS and SRNS patients. Our data not only illustrate the superiority of RTX in achieving relapse-free intervals but also show the unique benefits of the long-term administration of single RTX doses. Importantly, in patients with SRNS, repeated RTX dosing, when the initial treatment proved inadequate, resulted in delayed positive effects, indicating the drug’s capacity to modulate the immune response over time.

Historically, steroids have been the first-line treatment for NS, effectively inducing remission but often at the cost of significant relapse rates, particularly in SSNS patients, where nearly 50% of patients face frequent relapses [23]. The introduction of RTX has shifted the treatment paradigm, particularly in patients for whom steroids or CNIs were previously the standard treatment [24]. Numerous randomized trials, including ours, underscore that RTX treatment has comparable efficacy to traditional therapies but also reduces their associated toxicities [25,26,27,28,29].

Moreover, previous randomized clinical trials have revealed similar efficacy profiles among various steroid-sparing agents [2,23,29]. This underscores the significance of exploring alternative therapeutic options that could potentially offer comparable benefits while minimizing the adverse effects associated with long-term steroid use. The paradigm shifts towards biological therapies and the exploration of diverse treatment modalities contribute to the ongoing efforts to enhance the management of NS and glomerular diseases [30].

The use of monoclonal antibodies to target CD20 revolutionized the therapeutic approach to SRNS treatment, presenting advantages over previously used medications. This groundbreaking discovery suggested the significant role of B cells in idiopathic NS, leading to further studies being performed to characterize B-cell subpopulations in NS [27]. The direct interaction of RTX with acid sphingomyelinase-like phosphodiesterase 3b, present on podocytes, plays a crucial role in stabilizing their function and structure, thereby preventing NS relapse [30]. The use of RTX offers hope in terms of addressing the challenging issue of treating SRNS, particularly in pediatric nephrology [19,31].

The efficacy of anti-CD20 monoclonal antibodies in SRNS stands out as a noteworthy discovery from the last two decades [32]. Our study results aligned with previous research, demonstrating that RTX provides sustained remission in pediatric NS patients, even when used in challenging settings with limited monitoring capacity. For instance, Basu et al. reported that while RTX effectively induced remission, relapses typically occurred 12–18 months after B-cell recovery, necessitating re-administration [21]. Furthermore, our findings parallel those of studies suggesting that long-term RTX administration every 6–9 months, guided by CD20 levels, can effectively prevent relapses in SSNS and SRNS.

The economic considerations of RTX therapy, particularly its cost, remain a major discussion point. While the upfront expense of RTX might seem prohibitive, our preliminary analysis reveals that the long-term costs may be comparable to, or even lower than, those of the extended use of MMF or CNIs when factoring in hospitalizations and steroid-related complications [1,11,32]. This adds weight to the argument for using RTX, especially in resource-constrained settings where traditional immunosuppressive drug monitoring is impractical. The possibility of using RTX as an early therapeutic option during NS relapses or in progressive glomerulopathies offers an opportunity to streamline therapy without sacrificing efficacy [32]. 

While RTX treatment offers undeniable advantages, certain issues remain unaddressed, including the lack of means of definitively determining the final cumulative dose [33]. Recent studies comparing the efficacy of different courses of RTX highlight the potential significance of the cumulative dose, with patients receiving RTX after their third or fourth episode of NS relapse experiencing longer remissions than those treated after their first relapse [6,32,33]. This underscores the importance of ongoing research seeking to refine and optimize the use of RTX therapy in the management of SRNS.

In our study, there was a 72.2% complete/partial remission rate, suggesting the modifiability of the immune system and potentially excluding the possibility of NS having a purely genetic nature. Further in-depth studies are warranted to unravel the mechanisms of treatment and treatment options for children with SRNS. The genetic underpinnings of SRNS further complicate treatment, as steroid and cytostatic therapies prove ineffective in cases of genetically determined NS. Our results indicate that, even in patients with genetic forms of NS, RTX has the potential to modify the disease’s course, as seen in one patient who achieved remission after two years of RTX therapy, although kidney function declined later. The possibility of tailoring treatment based on genetic findings holds promise for personalized medicine approaches in NS [34]. Alternatively, kidney biopsy may be necessary in atypical cases or when genetic screening is unavailable. 

Safety remains a cornerstone of RTX therapy [1,11,33,34]. Our study observed a low to moderate incidence of adverse reactions, confirming RTX’s tolerability and supporting its continued use in pediatric populations. This aligns with other reports of RTX being tolerated well, even in cases of long-term use. Consideration should be given to desensitization protocols in order to manage allergic reactions to rituximab. In cases where standard regimens fail to control such reactions, we initiated the use of omalizumab. Although these results were not part of the current study, we found that administering 150 mg of omalizumab was highly effective in enabling the safe use of rituximab across various clinical scenarios. This approach significantly enhances tolerance and almost universally permits the continued administration of rituximab.

Despite the success of RTX therapy in terms of achieving and maintaining remission, caution is advised regarding repeated administration, especially at half doses, due to the heightened risk of infectious complications. Our findings suggest that maintenance RTX therapy should be avoided when IgG levels are below the age-appropriate range. Low IgG levels increase susceptibility to infections, even in the absence of clinical symptoms. Therefore, we emphasize the importance of withholding RTX re-administration in patients with low IgG levels, as this could prevent potential immunodeficiency-related infections. It is critical to closely monitor immunoglobulin levels and individualize therapy to balance efficacy with safety.

However, the optimal duration of RTX therapy remains an open question, with considerations of B-cell repopulation rates and CD20 monitoring influencing decisions on re-administration. Our experience supports the safety of extending the duration of RTX therapy to two years [20], but further studies are needed to confirm the long-term benefits and to refine dosing intervals.

In challenging conditions, such as those experienced during martial law in Ukraine, RTX has proven to be a viable first-line therapy [35]. Given the logistical hurdles associated with maintaining immunosuppressive regimens and monitoring pharmacokinetics in such contexts, RTX’s ability to maintain efficacy without frequent monitoring positions it as an attractive option [36]. Its potential to be used as a therapy of choice in low-resource settings further underscores the versatility of this biologic agent in pediatric nephrology [37]. 

Finally, the integration of RTX into the principles of “green nephrology”—aimed at optimizing resource use while ensuring effective care—is worth noting. RTX’s ability to reduce the need for multiple medications, decrease hospitalization rates, and streamline treatment offers an economically viable and sustainable solution for managing NS, particularly in challenging environments. This approach aligns with modern healthcare goals of achieving therapeutic efficacy with minimal environmental and economic impacts. This alignment with green nephrology principles underscores the importance of integrating advanced diagnostic techniques to enhance both clinical and economic aspects of green nephrology care in modern ecoreality. The consideration of RTX as part of green nephrology suggests a conscientious and resource-efficient approach to nephrological therapy. Despite the cost-related challenges, the potential benefits in terms of treatment efficacy, reduced reliance on other medications, and improved patient outcomes contribute to the broader discussion on sustainable and economically viable practices in the field of pediatric nephrology.

The concept of green nephrology embodies a prudent and economical approach to managing medical resources within the field of nephrology, aiming to achieve therapeutic effectiveness without compromising patient outcomes [38]. This work explores the incorporation of RTX as a potential component of green nephrology. Despite its initial high cost, the utilization of RTX demonstrates a commitment to resource efficiency. This is achieved through meticulous patient care, a decrease in the reliance on alternative treatments, a reduction in physicians’ observation time, and improved outcomes. The signs and characteristics observed in the application of RTX as part of green nephrology underscore its potential to contribute to a sustainable and economically viable approach to nephrological therapy.

## 5. Conclusions

The implementation of rituximab demonstrates substantial efficacy, with 100% remission achieved in steroid-sensitive nephrotic syndrome and sustained remission rates of 72.2% at seen 2 years in steroid-resistant nephrotic syndrome NS. Notably, a 2-year maintenance regimen of RTX has been shown to provide a relapse-free course for use in NS treatment. However, its therapeutic scope appears more limited in genetically determined nephrotic syndromes (GRNS), necessitating a more nuanced approach in such cases.

The findings emphasize the importance of combining efficacy with tolerability, reinforcing a patient-centered approach to care. RTX has been proven to streamline treatment protocols by reducing the adverse effects associated with conventional immunosuppressants while improving long-term therapy outcomes. Importantly, the careful administration of maintenance RTX doses is essential, particularly for the depletion of CD20-positive B cells. These doses should only be administered when IgG levels are within normal ranges and in the absence of clinically significant infections in order to mitigate the risk of immune-related complications. The side effects might be compared with those of other therapies with calcineurin inhibitors and mycophenolic acid in the discussion to classify them.

RTX emerges as a compelling alternative to traditional immunosuppressive therapies in NS, offering both clinical and economic advantages. Nonetheless, further investigation is required to optimize dosing strategies and fully explore RTX’s potential, particularly in GRNS and resource-constrained settings, where therapeutic options are limited.

The results of the RICHNESS add to the growing evidence base for RTX’s role in transforming the management of pediatric NS. Ongoing and further studies on the efficacy and safety of RTX, in children with NS are anticipated to yield evidence-based practice recommendations, guiding clinicians in terms of optimizing treatment strategies for this patient population and establishing definitive outcomes.

## Data Availability

Data supporting reported results can be found in clinic reports. Data are available on request from the authors.

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
