# Peer review of "Rituximab Administration to Treat Nephrotic Syndrome in Children: 2-Year Follow-Up"

_biomedicines, 2024, doi:10.3390/biomedicines12112600_

Round 1
Reviewer 1 Report
Comments and Suggestions for Authors
The authors evaluated the efficacy and safety of rituximab (RTX) in children with steroid-sensitive nephrotic syndrome (SSNS) and steroid-resistant nephrotic syndrome (SRNS) over a two-year period. However, the manuscript requires several improvements before it can be considered for publication in Biomedicines. Some key issues are outlined below:
(1) The manuscript is poorly formatted and does not adhere to the journal's guidelines, which affects the readability and clarity of some results.
(2) Some tables referenced in the text, such as Tables 4 and 5, are absent from the manuscript.
(3) Figure captions are inadequately described and lack important information necessary for proper interpretation.
(4) It is recommended that the methodology section be divided into subsections to improve its clarity and ease of understanding.
(5) The authors missed opportunities to apply statistical tools, such as mean comparison tests between continuous variables in the two experimental groups (SSNS and SRNS), as well as tests for assessing associations between categorical data.
(6) The abstract is overly long and needs to be rewritten for conciseness.
(7) The description of the statistical analysis is inconsistent with the statistical tests reported in the results section. The authors should report only the statistical tests that were actually used for the analysis and construction of the results in the methodology section.
Author Response
Dear Reviewer, we express our deep gratitude for reviewing our work, and let me respond to your comments, please.
Comment 1 The manuscript is poorly formatted and does not adhere to the journal's guidelines, which affects the readability and clarity of some results.
Response: Thank you for highlighting these issues. We acknowledge the importance of proper formatting for clarity and readability. We ensure that the manuscript adheres to the journal's guidelines and improved the presentation of our results. Furthermore, we appreciate your feedback and have made the necessary revisions.
Comment 2 Some tables referenced in the text, such as Tables 4 and 5, are absent from the manuscript.
Response. These two tables are placed at the end of the article and are now attached to the article as a separate file, since they are horizontal due to the large amount of data. Please excuse their incorrect placement.
Comment 3. Figure captions are inadequately described and lack important information necessary for proper interpretation.
Response. Thank you for pointing that out. We revised and added notes to Figure 2 and graphs 1,2 to ensure they are thoroughly described and include all necessary information for clear and accurate interpretation. Your feedback is invaluable in improving the clarity of our manuscript.
Comment 4. It is recommended that the methodology section be divided into subsections to improve its clarity and ease of understanding.
Response: Thank you for the suggestion. We divided the methodology section into subsections to enhance clarity and make it easier to follow. Your feedback is appreciated and were implemented to improve the overall readability of the manuscript.
Comment 5. The authors missed opportunities to apply statistical tools, such as mean comparison tests between continuous variables in the two experimental groups (SSNS and SRNS), as well as tests for assessing associations between categorical data.
Response: We appreciate your keen observation. We acknowledge the importance of robust statistical analysis to enhance the validity of our findings. We incorporated mean comparison tests between continuous variables in the SSNS and SRNS groups and also included tests for assessing associations between categorical data (tables 4-5). Your feedback is invaluable and was implemented to strengthen our manuscript.
Comment 6. The abstract is overly long and needs to be rewritten for conciseness.
Response: Thank you for your input. We revised the abstract to ensure it is more concise while retaining all critical information. Your feedback is crucial for refining the manuscript, and we appreciate your attention to detail.
Comments 7. The description of the statistical analysis is inconsistent with the statistical tests reported in the results section. The authors should report only the statistical tests that were actually used for the analysis and construction of the results in the methodology section.
Response: Thank you for pointing this out. We revised the methodology section to ensure it accurately reflects the statistical tests that have been used in the analysis and the construction of the results. Your feedback is vital for maintaining the integrity and clarity of our work.
Thank you once again for your insightful comments. We deeply appreciate the time and effort you have dedicated to reviewing our manuscript. Your feedback has been invaluable in enhancing the quality and clarity of our work.
With sincere thanks, D. Ivanov

Reviewer 2 Report
Comments and Suggestions for Authors
Dear authors,
Congratulation on your excellent work! It was a pleasure reading your well-organized and skillfully written article. I suggest a few changes that I believe the manuscript would benefit from.
1. Please consider presenting rituximab’s mechanisms of action that explain its efficacy in SSNS and SRNS. A diagram illustrating these mechanisms would make the information more accessible.
2. I recommend including a brief discussion on potential biomarkers predictive of response to rituximab in SSNS and SRNS patients.
3. “Low IgG increases susceptibility to infections, even in the absence of clinical symptoms. Therefore, we emphasize the importance of withholding RTX re-administration in patients with low IgG, as this could prevent potential immunodeficiency-related infections. It is critical to closely monitor immunoglobulin levels and 400 individualize therapy to balance efficacy with safety.” (397-401) Adjunctive intravenous immunoglobulins could be used in this setting. Please consider commenting on the association of IgIV to RTX treatment.
4. Desensitisation to rituximab should also be discussed considering that 2 of the patients initially included in the study did not benefit from RTX treatment following the development of allergic reactions.
Best regards!
Author Response
Dear Reviewer, we express our deep gratitude for reviewing our work, its high assessment and very important comments and wishes. Allow me to respond to your comments.
Comment 1 Please consider presenting rituximab’s mechanisms of action that explain its efficacy in SSNS and SRNS. A diagram illustrating these mechanisms would make the information more accessible.
Response: We have summarized the mechanisms of action and presented it in the form of a graph, placing it on lines 93-95. You are absolutely right, Your suggestion improves reader comprehension and justifies our use of rituximab.
Comment 2 I recommend including a brief discussion on potential biomarkers predictive of response to rituximab in SSNS and SRNS patients.
Response. We added a brief discussion with recent literature (lines 96-110), citing sources from mainly 2023-2024 (Iijima et al., Beck et al., Ravani et al., Larkins & Gipson, and Mishra & Kalinina) – references 13-17.
Comment 3. Low IgG increases susceptibility to infections, even in the absence of clinical symptoms. Therefore, we emphasize the importance of withholding RTX re-administration in patients with low IgG, as this could prevent potential immunodeficiency-related infections. It is critical to closely monitor immunoglobulin levels and individualize therapy to balance efficacy with safety.” (397-401) Adjunctive intravenous immunoglobulins could be used in this setting. Please consider commenting on the association of IgIV to RTX treatment.
Response. Yes, we did perform such correction in 4 (57%) of 7 children with detected hypogammaglobulinemia. I missed this fact and added these data to the obtained results (lines 349-350). Thank you for highlighting this.
Comment 4. Desensitisation to rituximab should also be discussed, considering that 2 of the patients initially included in the study did not benefit from RTX treatment following the development of allergic reactions.
Response: This remark is very correct for practice. We did not conduct classical desensitization, but in two children who had such a reaction to the introduction of rituximab that the standard protocol did not allow overcoming, we delayed the iv of RTX for 2 days, used 150 mg of omalizumab and the next day successfully re-administrated rituximab. These children were not included in the study. We included a note in the discussion (lines 422-428). Thank you kindly.
We greatly appreciate your comments, which have strengthened the manuscript, thank you once more.
Yours Sincerely, D. Ivanov
Round 2
Reviewer 1 Report
Comments and Suggestions for Authors
The submitted version does not include the changes highlighted, which makes it difficult to accurately assess the adjustments made by the authors. Additionally, the manuscript contains some issues related to the journal's author guidelines that need to be addressed before acceptance. It is unclear why the authors had difficulty properly formatting Tables 4 and 5 to ensure their correct placement in the manuscript. The text also contains some poorly structured paragraphs, composed of only one sentence, and has formatting issues (e.g., incorrect use of superscripts)."
Author Response
Dear Reviewer,
Thank you for your valuable feedback. I sincerely apologize for the issues in the current version of the manuscript. We understand the importance of highlighting changes and have now ensured that all significant revisions are clearly marked (with red). Additionally, we have reformatted Tables 4 and 5 according to the journal’s guidelines to ensure proper placement. We also addressed the structural issues within the paragraphs, improving sentence flow and correcting formatting errors like superscripts. I hope these updates meet the journal's standards.
Best regards,
D. Ivanov

Round 3
Reviewer 1 Report
Comments and Suggestions for Authors
Despite of some changes and correction, the manuscript is not formatted according the authors rules and have some important errors linked trivial points, such as use the term “graph” in the place of the term “figure”, tables and paragraphs spacing are poorly diagramed.
Author Response
Dear Reviewer,
Thank you for your detailed feedback on our manuscript. We appreciate the time and effort you have taken to highlight areas for improvement. We apologize for the oversight regarding formatting and trivial errors. We have carefully revised the manuscript to ensure it aligns with the author guidelines. Specifically, we have replaced the term “graph” with “figure” where appropriate, adjusted the spacing and layout of tables and paragraphs, and thoroughly reviewed the document for any other formatting inconsistencies.
Additionally, The tables 4 and 5 are attached separately due the large format.
We are committed to maintaining a high standard and are grateful for your guidance in helping us improve the quality of our work. Please let us know if there are any additional adjustments needed.
Sincerely, D.Ivanov
